



# Changing mineralogical properties of shells may help minimize the impact of hypoxia-induced metabolic depression on calcification

Jonathan Y.S. Leung[1,2], Napo K.M. Cheung[2,3]

[1] School of Biological Sciences, The University of Adelaide, Adelaide, Australia

[2] Department of Biology and Chemistry, City University of Hong Kong, Hong Kong SAR

[3] Department of Biological Sciences, Graduate School of Science, The University of Tokyo, Tokyo, Japan

*Correspondence to*: Jonathan Y.S. Leung (jonathan_0919@hotmail.com)

**Abstract.** The occurrence of hypoxia becomes more prevalent in coastal and marine waters due to ocean warming and human-induced eutrophication. While hypoxia is expected to hamper calcification via metabolic depression, recent studies showed that some calcifying organisms can maintain normal shell growth. The underlying mechanism is unclear, but may be associated with energy reallocation or mineralogical plasticity which reduces the energy demand for calcification. We tested the hypothesis that shell growth can be maintained under hypoxia by compromising the mechanical strength of shells as the trade-off, or changing the mineralogical properties of shells. The respiration rate, clearance rate, shell growth rate and shell properties of a calcifying polychaete (*Hydroides diramphus*) were determined under normoxia or hypoxia in two contexts (life-threatening and unthreatened conditions). Despite the reduced respiration rate and clearance rate under hypoxia, harder and stiffer shells were still produced at a higher rate under life-threatening conditions. The maintenance of this anti-predator response is possibly attributed to the reduced energy demand for calcification by altering mineralogical properties (e.g. increased calcite to aragonite ratio). Our findings suggest that this compensatory mechanism may enable calcifying organisms to maintain calcification under hypoxia and acclimate to the future metabolically stressful environment.

## 1 Introduction

Anthropogenic activities have caused drastic changes in marine ecosystems over the last century. For instance, the accelerated emission of carbon dioxide since the onset of Industrial Revolution has already led to pH reduction in seawater, known as ocean acidification (Doney et al., 2009; IPCC, 2013). On the other hand, the occurrence of hypoxia (dissolved oxygen (DO) concentration $\leq 2.0$ mg $O_2$ $L^{-1}$) becomes more prevalent in marine ecosystems due to human-induced eutrophication and ocean warming which decreases the solubility of oxygen in seawater (Diaz and Rosenberg, 2008; Keeling et al., 2010; Bijma et al., 2013). Ocean acidification and hypoxia are regarded as the major threats to the fitness and survival of marine organisms, possibly causing serious ramifications on marine ecosystems (Bijma et al., 2013).



Calcifying organisms (e.g. corals, molluscs, polychaetes and echinoderms) are very diverse in marine ecosystems, but they are considered to be particularly susceptible to acidifying oceans where the reduced carbonate saturation state can retard shell growth (or calcification) (Doney et al., 2009). However, this proposition remains

controversial as many calcifying organisms can maintain or even enhance calcification under $CO_2$-induced ocean acidification (Ries et al., 2009; Leung et al., 2017). It has been suggested that carbonate saturation state does not necessarily influence calcification since calcifying organisms may be able to use bicarbonate ions for this process (Ries, 2011a; Bach, 2015). Instead, growing evidence reveals that calcification is mainly driven by the energy metabolism of calcifying organisms since this process is energy-dependent (Findlay et al., 2009; Wijgerde et al., 2014).

From the energetic perspective, therefore, hypoxia can pose similar impacts as ocean acidification on calcification through impaired energy metabolism (Cheung et al., 2008; Wijgerde et al., 2014).

Yet, some calcifying organisms have been shown to maintain calcification under hypoxia (Mukherjee et al., 2013; Nardelli et al., 2014; Keppel et al., 2016). It is favourable because shell growth is vital for not only continuing somatic growth, but also offering physical protection, especially under life-threatening conditions (e.g. harder shells

produced at a higher rate under predation risk) (Cheung et al., 2004; Brookes and Rochette, 2007; Hirsch et al., 2014). Given the impaired energy metabolism, the sustained shell growth under hypoxia suggests potential mechanisms which compensate for the reduced metabolic energy. It may be achieved by energy reallocation where the metabolic energy for calcification is maintained at the expense of other physiological processes (Sokolova et al., 2012). Alternatively, the energy demand for calcification can be modified by altering the mineralogical properties of shells,

such as carbonate polymorphs (Weiner and Addadi, 1997, 2011; Jacob et al., 2008), potentially offsetting the reduced energy budget under hypoxia. Whether calcifying organisms can counter the impacts of hypoxia on shell growth by reallocating metabolic energy or modifying energy demand remains unexplored and deserves a comprehensive investigation.

To unravel the potential compensatory mechanisms to hypoxia, we selected a calcifying polychaete as a

55 model species, which is tolerant and resilient to hypoxia (Vaquer-Sunyer and Duarte, 2008; Leung et al., 2013). Despite the possible reduction in energy metabolism and energy intake by feeding under hypoxia, we hypothesized that (1) shell growth rate can be maintained, but the mechanical strength of shells (hardness and elasticity) is weakened as the trade-off; (2) the mineralogical properties of shells (calcite to aragonite ratio, magnesium to calcium ratio in calcite and amorphous calcium carbonate content) are modified to reduce the energy demand for calcification. Provided

energy reallocation and mineralogical plasticity can act as compensatory mechanisms which allow the maintenance of shell growth under hypoxia, this not only substantiates that calcification is mainly subject to the energetics of calcifying organisms, but also suggests that calcifying organisms may be more capable of acclimating to metabolically stressful conditions, such as ocean acidification and hypoxia, than previously thought.

**2 Materials and methods**

**2.1 Experimental design**



A calcifying polychaete *Hydroides diramphus*, which has a widespread distribution within circumtropical regions (Çinar, 2006), was chosen as the study species. Based on our preliminary study, this tube-building polychaete was tolerant to hypoxia and able to maintain survival above 1.5 mg $O_2$ $L^{-1}$. Individuals were collected from a fish farm at

70 Yung Shue O (22°25′N, 114°16′E), Hong Kong, in summer when hypoxia was observed (Leung et al., 2013). In the laboratory, they were carefully separated from other fouling organisms (e.g. mussels and tunicates) and then kept in plastic aquaria (50 cm × 40 cm × 30 cm) with natural seawater under laboratory conditions (temperature: $28.0 \pm 1.0$°C, DO: $6.00 \pm 0.10$ mg $O_2$ $L^{-1}$, salinity: $33.0 \pm 0.5$ ppt and pH: $8.10 \pm 0.05$). An algal suspension containing *Isochrysis galbana* and *Dunaliella tertiolecta* (1:1, v/v) was daily provided as food. Individuals were allowed to acclimate for

one week prior to experimentation.

In this study, two DO levels were chosen to represent normoxia (6.0 mg $O_2$ $L^{-1}$) and hypoxia (2.0 mg $O_2$ $L^{-1}$). Normoxia was maintained by aerating seawater with air, whereas hypoxia was achieved by aerating seawater with a mixture of nitrogen and air. The flow rate of each gas was regulated by a digital flow meter (Vögtlin Instruments, Switzerland) and the DO concentration of seawater was monitored by the Stable Optical Oxygen System (SOO-100, TauTheta Instruments, USA). In the following experiments, seawater temperature was maintained at 28°C using a

80 heating bath circulator to simulate the seawater temperature at the collection site in summer.

The effect of DO was examined in two contexts: life-threatening and unthreatened (i.e. control) conditions. The former was triggered by carefully trimming the calcareous tube until the radioles were exposed, while the body was still fully covered. This mimics the non-lethal shell damage after being attacked by predators. The individuals

with "intact" (tube length: ~40 mm) and "damaged" (tube length: ~20 mm) tubes were allowed to acclimate under either normoxia or hypoxia for another week prior to experimentation.

### 2.2 Shell growth and shell properties

To determine shell growth, individuals with their tube length measured were separately transferred into a 2 ml microcentrifuge tube with the radioles pointing upward ($n$ = 10 individuals per replicate). A small hole was drilled at

90 the bottom of each microcentrifuge tube to allow water exchange. The microcentrifuge tubes were glued together to maintain an upright position and then put into a glass bottle containing 180 ml filtered seawater (FSW) (pore size: 45 µm) with DO level manipulated ($n$ = 3 replicate bottles per DO level per context). Food was daily provided by adding 20 ml algal suspension containing *I. galbana* and *D. tertiolecta* (1:1, v/v) at ~$1 \times 10^6$ cells $ml^{-1}$. The microcentrifuge tubes were cleaned and seawater was renewed once every three days to prevent accumulation of metabolic waste.

After three weeks, the tube length of each individual was measured to estimate shell growth.

Following the measurements of respiration rate and clearance rate (see section 2.3), the newly-produced shells were carefully removed for the analyses of mechanical and geochemical properties. Given the limited amount, shells (ca. 3 – 5 individuals) from the same treatment were pooled to make a composite sample.

The hardness and elasticity of shells were measured using a micro-hardness tester (Fischerscope HM2000,

Fischer, Germany). A shell fragment was mounted firmly onto a metal disc using cyanoacrylate adhesives ($n$ = 5 fragments from 5 individuals per DO level per context). Then, the fragment was indented by a Vickers 4-sided diamond





pyramid indenter for 10 s in the loading phase (Peak load: 300 mN; Creep: 2 s). In the unloading phase, the load decreased at the same rate as the loading phase until the loading force became zero. At least five random locations on each fragment were indented. Vickers hardness and elastic modulus were calculated based on the load-displacement curve using software WIN-HCU (Fischer, Germany).

Carbonate polymorphs were analysed using an X-ray diffractometer (D4 ENDEAVOR, Bruker, Germany). A small quantity of shell powder was transferred onto a tailor-made sample holder and then scanned by Co Kα radiation (35 kV and 30 mA) from 20° to 70° 2θ with step size of 0.018° and step time of 1 s ($n$ = 3 replicates per DO level per context). Carbonate polymorphs were identified based on the X-ray diffraction spectrum using the EVA XRD analysis software (Bruker, Germany). Calcite to aragonite ratio was estimated using the following equation (Kontoyannis and Vagenas, 2000):

$$\frac{I_C^{104}}{I_A^{221}} = 3.157 \times \frac{X_C}{X_A}$$

where $I_C^{104}$ and $I_A^{221}$ are intensity of the calcite 104 peak (34.4° 2θ) and aragonite 221 peak (54.0° 2θ), respectively; $X_C/X_A$ is the calcite to aragonite ratio.

Magnesium to calcium ratio was determined by energy dispersive X-ray spectroscopy under the Philips XL 30 field emission scanning electron microscope. A small quantity of shell powder was transferred onto a stub and coated by carbon ($n$ = 3 replicates per DO level per context; 3 trials per replicate). The shell powder was irradiated by an electron beam with an accelerating voltage of 12 kV to obtain the energy spectrum with background correction. Elements were identified and magnesium to calcium ratio was calculated using software Genesis Spectrum SEM Quant ZAF (EDAX, USA).

To determine relative amorphous calcium carbonate (ACC) content, 1 mg shell powder was mixed with 10 mg potassium bromide, followed by compressing the mixture into a disc using a manual hydraulic press. An infrared absorption spectrum ranging from 400 cm⁻¹ to 4000 cm⁻¹ with background calibration for the baseline was obtained using a Fourier transform infrared spectrometer (Avatar 370 DTGS, Nicolet, USA) ($n$ = 3 replicates per DO level per context). The relative ACC content was estimated as the intensity ratio of the peak at 856 cm⁻¹ to that at 713 cm⁻¹ (Beniash et al., 1997).

### 2.3 Physiological performance

Respiration rate and feeding rate were measured following the 3-week exposure period. To measure respiration rate, five individuals were transferred into a syringe containing ~35 ml FSW with DO level manipulated ($n$ = 5 replicate syringes per DO level per context). They were allowed to acclimate in the syringe for 15 min, followed by expelling the air inside and sealing the syringe for one hour. The initial and final DO concentrations of FSW in the syringe were measured using an optical dissolved oxygen probe (SOO-100, TauTheta Instruments, USA). Blank samples without individuals were prepared to correct the background change in DO concentration.



To estimate clearance rate, five individuals which had been starved for one day to standardize their hunger
level were transferred into a glass bottle containing 80 ml FSW with an initial concentration of ~$1 \times 10^6$ cell ml$^{-1}$ *D. tertiolecta* ($n = 5$ replicate bottles per DO level per context). After feeding for one hour, 1 ml seawater was taken from the bottle and the microalgae were enumerated using a haemocytometer ($n = 6$ trials per bottle). Prior to counting, 1% Lugol's solution was used to fix the microalgae. Clearance rate was calculated using the following formula (Coughlan, 1969):

$$CR = \frac{V}{nt} \times ln\frac{C_o}{C_t}$$

where *CR* is the clearance rate (ml ind$^{-1}$ hr$^{-1}$); *V* is the volume of seawater; *n* is the number of individuals; *t* is the feeding time; $C_o$ and $C_t$ are the initial and final algal concentrations, respectively.

**2.4 Statistical analysis**

Two-way permutational analysis of variance (PERMANOVA) was conducted to test the effect of DO and context on the aforementioned parameters using software PRIMER 6 with PERMANOVA+ add-on.

**3 Results**

The shell growth of *H. diramphus* was significantly faster following non-lethal shell damage, while hypoxia only slightly retarded shell growth (Fig. 1a, Table 2). Harder and stiffer (i.e. higher elastic modulus) shells were produced following non-lethal shell damage, whereas the effect of hypoxia on hardness and elasticity was negligible (Figs. 1b and 1c, Table 2).

Respiration rate was downregulated substantially by hypoxia, but only slightly by non-lethal shell damage (Tables 1 and 2). Clearance rate was reduced under hypoxia, regardless of context (Tables 1 and 2). However, non-lethal shell damage led to a significant reduction in clearance rate under normoxia. Regarding the geochemical properties, calcite was the dominant carbonate polymorph and its proportion was elevated under hypoxia (Tables 1 and 2). High-Mg calcite (i.e. Mg/Ca > 0.04) was produced by *H. diramphus* and the Mg content in calcite increased under hypoxia (Tables 1 and 2). Hypoxia led to the production of less crystalline shells, indicated by the higher relative ACC content (Tables 1 and 2). All the geochemical properties were not significantly influenced by non-lethal shell damage.

**4 Discussion**

While hypoxia has been regarded as a major threat to marine organisms (Wu, 2002; Diaz and Rosenberg, 2008), many less mobile marine taxa (e.g. molluscs, annelids and echinoderms) are generally tolerant to it (Vaquer-Sunyer and Duarte, 2008), implying their potential capacity to accommodate its impacts. Despite the reduced energy metabolism and energy intake by feeding, we demonstrated that a calcifying polychaete is able to maintain shell growth and





mechanical strength under hypoxia probably by reallocating energy budget (i.e. energy trade-offs between shell quality and shell quantity) and changing mineralogical properties, which have adaptive values.

As calcification is an energy-demanding process (Palmer, 1992), it is predicted that the quality and quantity of shells produced by calcifying organisms would be worsened under hypoxia due to its impacts on energy intake and
metabolism (Cheung et al., 2008; Leung et al., 2013; Wijgerde et al., 2014). Under unthreatened conditions (i.e. without non-lethal shell damage), we observed that the shell growth of *H. diramphus* is retarded but the mechanical strength of new shells is maintained under hypoxia. The slower shell growth could be associated with the reduced feeding rate, which leads to reduced energy budget for calcification. Indeed, some studies suggest that energy intake by feeding is the key for maintaining shell growth (Melzner et al., 2011; Thomsen et al., 2013; Leung et al., 2017).
However, energy reserves can only be efficiently converted into metabolic energy via aerobic respiration. In this regard, the slower shell growth is more likely attributed to the hypoxia-induced metabolic depression, which limits the amount of metabolic energy for calcification. Contrary to our prediction, both hardness and elasticity of shells were unaffected by hypoxia. These mechanical properties are mainly determined by the quantity of matrix proteins occluded in the shell, which can exceptionally augment the mechanical strength (Weiner and Addadi, 1997; Addadi et al., 2006; Marin
et al., 2008). Our findings, therefore, imply that similar amount of metabolic energy can still be allocated to the production of matrix proteins under hypoxia (Mukherjee et al., 2013), so that the mechanical strength can be maintained. When energy budget becomes limited, this energy allocation strategy (i.e. shell quality over shell quantity) is adaptive because there is no exigency to expedite shell growth when stress is not imminent and protection provided by the shell is adequate.

Under life-threatening conditions (i.e. following non-lethal shell damage), harder and stiffer shells were produced at a higher rate, despite the reduced energy intake by feeding. This is a typical anti-predator response since shell repair should be prioritized to restore and enhance protection (Cheung et al., 2004; Hirsch et al., 2013; Brom et al., 2015). This response can be achieved by downregulating the less essential physiological processes or activities as trade-offs (Rundle and Brönmark, 2001; Trussell and Nicklin, 2002; Hoverman and Relyea, 2009; Babarro et al.,
2016). For example, Brookes and Rochette (2007) revealed that gastropod *Littorina obtusata* is able to actively increase calcification rate under predation risk at the expense of grazing activity and somatic growth. Indeed, anti-predator response should be prioritized to maximize the chance of survival, when survival is not guaranteed (Bourdeau, 2009). Given the immobility of *H. diramphus*, the only effective anti-predator response is producing harder shells at a higher rate. In addition to a possible increase in the production of matrix proteins, the greater shell strength may be
due to the formation of microstructures in the shell (Meyers et al., 2008; Brom et al., 2015), which warrants further investigation.

Given the enormous energy demand for calcification and production of matrix proteins (Palmer, 1992), however, it is likely that the anti-predator response would be compromised under hypoxia due to the reduced energy budget. Contrary to this prediction, mechanical strength was maintained and shell growth was only slightly retarded,
meaning that the anti-predator response is not markedly affected by hypoxia. This unexpected observation not only indicates the strong tolerance of *H. diramphus* to hypoxia, but also suggests potential mechanisms allowing the



maintenance of shell quality and shell quantity produced under metabolic depression. While the maintenance of mechanical strength could be due to the production of relatively less energy-costly microstructural layers (Bourdeau, 2009), the rapid shell growth still requires substantial metabolic energy. Therefore, solely changing energy allocation strategy is unlikely to enable the maintenance of shell growth under hypoxia. We propose that changing mineralogical properties could help compensate for the reduced metabolic energy under hypoxia and hence allows the sustained shell growth. We found that *H. diramphus* consistently altered its mineralogical properties in response to hypoxia-induced metabolic depression. For example, an elevated proportion of calcite was precipitated under hypoxia, regardless of context. This strategy is favourable since precipitation of calcite is less energy-demanding than that of aragonite (Hautman, 2006; Ries, 2011b). In addition, precipitation of calcite allows faster shell growth due to lower density (Weiner and Addadi, 1997), which can partly explain why the shell growth was only slightly retarded under severe metabolic depression. Indeed, increased precipitation of calcite is an adaptive strategy for maintaining shell growth under metabolic depression (Ramajo et al., 2015; Leung et al., 2017). Amorphous calcium carbonate is the precursor of crystalline calcium carbonate which is formed following crystallization (Addadi et al., 2006). This biological process, where the unstable amorphous form is transformed into the stable crystalline form, requires metabolic energy as various proteins are involved in the transport of carbonate ions (Weiner and Addadi, 2011). We observed that crystallization is slightly hampered by hypoxia, indicated by the higher relative ACC content. This implies that metabolic energy allocated to crystallographic control is reduced, resulting in the production of shells of lower integrity (Fitzer et al., 2013). Magnesium is incorporated into the crystal lattice of calcite during calcification, affecting shell solubility and integrity (Fernandez-Diaz, 1996; Ries, 2011b). Some calcifying organisms are able to regulate the incorporation of magnesium into calcite to minimize its impacts (Nash et al., 2015). Although the underlying mechanism is still enigmatic, it is predicted that metabolic energy is involved in the control of magnesium incorporation. Our results imply that this control is undermined by hypoxia probably due to the reduced metabolic energy, thereby culminating in the production of calcite of higher Mg/Ca, which has higher solubility (Ries, 2011b).

Since hypoxia can be persistent, or even permanent, as observed in many coastal and marine waters worldwide (Helly and Levin, 2004; Diaz and Rosenberg, 2008), the survival of some calcifying organisms under persistent hypoxia suggests that they may have evolved the capacity to accommodate its impacts on energy metabolism and hence maintain calcification (Mukherjee et al., 2013; Frieder et al. ,2014; Nardelli et al., 2014; Keppel et al., 2016). Yet, the underlying mechanism remains unclear. While energy reallocation can possibly allow the maintenance of calcification under metabolic depression (Seibel, 2011; Sokolova et al., 2012), this strategy is considered to be maladaptive in the long term since calcifying organisms cannot achieve maximal physiological performance due to the trade-offs (Wu, 2002). In this regard, we propose that changing mineralogical properties, which minimizes the energy demand for calcification, is likely the overarching mechanism enabling calcifying organisms to maintain calcification under persistent hypoxia (see also Leung et al., 2017).

In conclusion, the occurrence of hypoxia in coastal and marine waters becomes more prevalent due to ocean warming and human-induced eutrophication; therefore, calcifying organisms will be more frequently impacted by hypoxia, unless they can exhibit compensatory mechanisms to minimize its impacts on calcification. Despite the



reduced energy budget, we demonstrated that hypoxia only slightly retards shell growth without weakening mechanical strength and anti-predator response can be maintained (i.e. harder shells produced at a higher rate). This is likely achieved by changing mineralogical properties of shells, which reduces the energy demand for calcification. Despite some trade-offs (e.g. shell solubility), this compensatory mechanism possibly helps calcifying organisms acclimate to metabolically stressful conditions, and hence sustain their populations and ecological functions in coastal and marine ecosystems.

*Acknowledgements.* Financial support was provided by the University Grants Committee of Hong Kong Special Administrative Region (AoE/P-04/04) and the IPRS Scholarship from the University of Adelaide to JYSL.

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



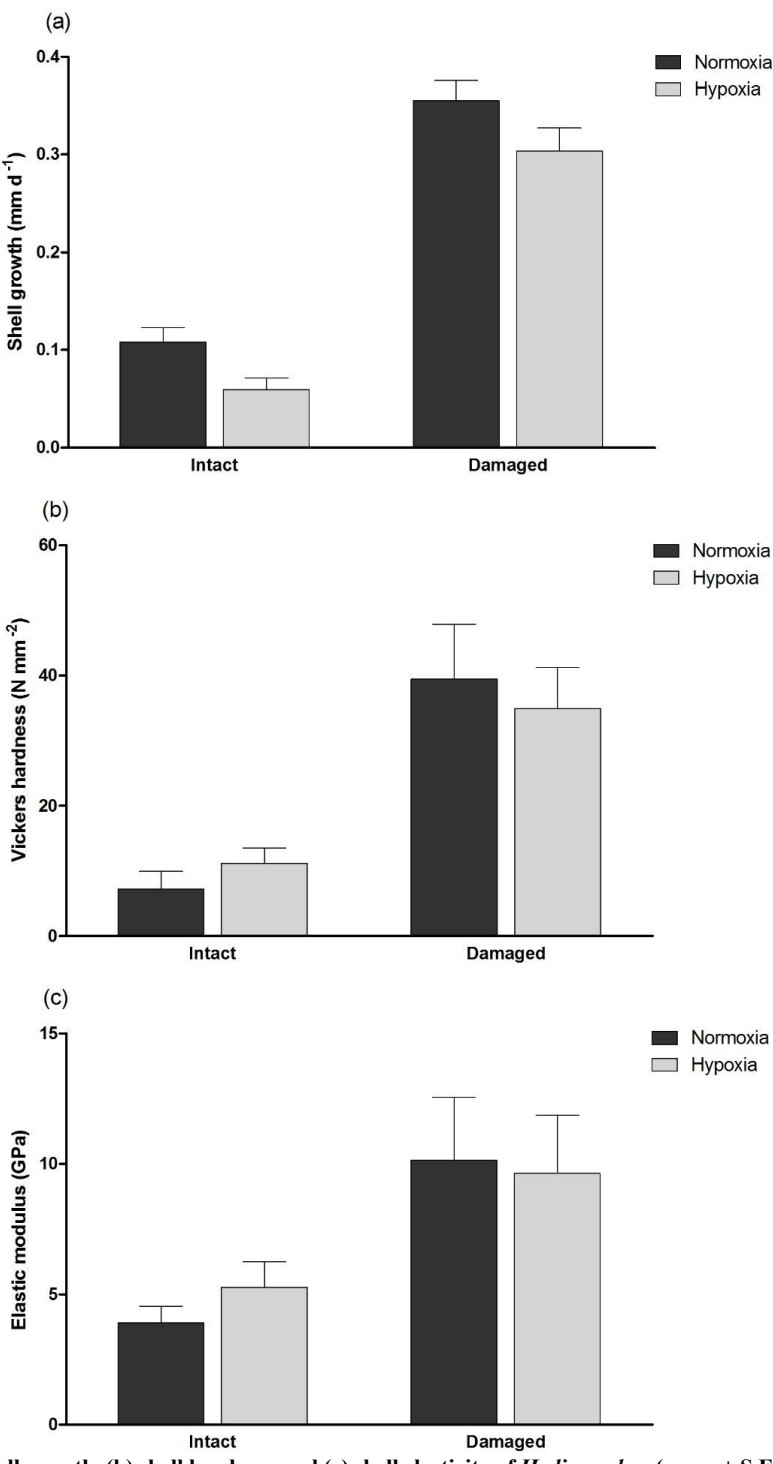

**Figure 1 (a) Shell growth, (b) shell hardness and (c) shell elasticity of *H. diramphus* (mean + S.E.), reflecting the quality and quantity of shells produced at different DO levels in different contexts.**



**Table 1 Physiological performance and shell geochemical properties of *H. diramphus* at different DO levels in different contexts (mean ± S.E.).**

|  | Intact | | Damaged | |
| --- | --- | --- | --- | --- |
|  | Normoxia | Hypoxia | Normoxia | Hypoxia |
| Physiological performance | | | | |
| Respiration rate ($\mu g$ $O_2$ $ind^{-1}$ $hr^{-1}$) | $26.6 \pm 0.22$ | $6.00 \pm 0.23$ | $25.5 \pm 0.51$ | $4.58 \pm 0.19$ |
| Clearance rate (ml $ind^{-1}$ $hr^{-1}$) | $9.33 \pm 0.55$ | $1.85 \pm 0.31$ | $1.72 \pm 0.19$ | $1.01 \pm 0.22$ |
| Geochemical properties | | | | |
| Calcite/aragonite | $0.731 \pm 0.057$ | $0.938 \pm 0.049$ | $0.766 \pm 0.040$ | $0.865 \pm 0.036$ |
| Mg/Ca in calcite | $0.154 \pm 0.014$ | $0.218 \pm 0.022$ | $0.126 \pm 0.014$ | $0.216 \pm 0.024$ |
| Relative ACC content | $2.65 \pm 0.04$ | $3.19 \pm 0.19$ | $2.58 \pm 0.08$ | $2.73 \pm 0.18$ |





**Table 2** PERMANOVA table showing the effect of DO and context on shell growth, Vickers hardness, elastic modulus,

respiration rate, clearance rate, calcite/aragonite, Mg/Ca in calcite and relative ACC content.

| | MS | F | *p* | Comparison of mean |
|---|---|---|---|---|
| Shell growth | | | | |
| DO | $7.59 \times 10^{-3}$ | 7.49 | **0.027** | Normoxia > Hypoxia |
| Context | 0.181 | 178 | **0.004** | Damaged > Intact |
| DO × Context | $7.76 \times 10^{-6}$ | $7.66 \times 10^{-3}$ | 0.892 | |
| Vickers hardness | | | | |
| DO | 0.293 | 1.90 | 0.974 | |
| Context | $3.92 \times 10^{3}$ | 25.5 | **0.001** | Damaged > Intact |
| DO × Context | 88.3 | 0.574 | 0.459 | |
| Elastic modulus | | | | |
| DO | 0.900 | 0.059 | 0.812 | |
| Context | 141 | 9.18 | **0.008** | Damaged > Intact |
| DO × Context | 4.37 | 0.285 | 0.601 | |
| Respiration rate | | | | |
| DO | $2.15 \times 10^{-3}$ | $4.36 \times 10^{3}$ | **0.001** | Normoxia > Hypoxia |
| Context | $8.52 \times 10^{-6}$ | 14.2 | **0.001** | Intact > Damaged |
| DO × Context | $7.40 \times 10^{-8}$ | 0.150 | 0.715 | |
| Clearance rate | | | | |
| DO | 84.0 | 140 | **0.001** | Within Intact: Normoxia > Hypoxia<br>Within Damaged: Normoxia > Hypoxia |
| Context | 89.1 | 148 | **0.001** | Within Normoxia: Intact > Damaged<br>Within Hypoxia: N.S. |
| DO × Context | 57.2 | 95.5 | **0.001** | |
| Calcite/Aragonite | | | | |
| DO | 0.070 | 11.0 | **0.017** | Hypoxia > Normoxia |
| Context | $1.14 \times 10^{-3}$ | 0.178 | 0.623 | |
| DO × Context | $8.84 \times 10^{-3}$ | 1.39 | 0.249 | |
| Mg/Ca in calcite | | | | |
| DO | 0.018 | 16.0 | **0.004** | Hypoxia > Normoxia |
| Context | $6.92 \times 10^{-4}$ | 0.618 | 0.455 | |
| DO × Context | $4.83 \times 10^{-4}$ | 0.431 | 0.530 | |
| Relative ACC content | | | | |
| DO | 0.355 | 6.02 | **0.047** | Hypoxia > Normoxia |
| Context | 0.199 | 3.37 | 0.104 | |
| DO × Context | 0.116 | 1.97 | 0.206 | |

357