# Peer review of "Changing mineralogical properties of shells may help minimize the impact of hypoxia-induced metabolic depression on calcification"

_Biogeosciences, 2017_

## Referee Comment (RC1) · Anonymous Referee #1 · 1 May 2017

In this paper, J. Leung and N. Cheung describe and discuss the effect of hypoxia on growth, shell properties and chemistry, respiration and feeding activity of Hydroides diramphus.

Overall the paper is well written, the data are well presented and discussed but one aspect of the experimental design concerns me. The authors achieved incubation in hypoxia by "aerating seawater with a mixture of nitrogen and air" (l. 76-77). This technique indeed allows replacement of dissolved oxygen by $N_2$ but also tends to remove dissolved $CO_2$ and can then increase water pH. Did the authors measure and control

pH in the different DO treatments? If yes, please provide this information. If no, I'm afraid that most of the discussion on the effect of hypoxia on shell chemistry might be irrelevant giving that changes in pH could explain the changes in calcite/aragonite and Mg/Ca. The lack of discussion of the effect of pH is especially surprising since you mention at least 6 times the ocean acidification in the introduction and none in the discussion. I would therefore recommend to further discuss the potential changes in pH during the experiment (ideally you should measure it) and how it could affect the calcification of this polychaete species.

To summarize, I would recommend the publication of the present paper in the following conditions:

-the authors detail if/how they controlled the pH during the experiment;

- estimate or measure the pH changes during N2 bubbling;

- discuss of how could pH changes (if existing) could explain together with hypoxia the shell chemistry observations you made.

In addition, here are some extra comments on the manuscript: Introduction section. In the introduction, you mention at least 6 times the ocean acidification, a phenomenon that you do not mention again at any point later in the paper. As explained above I think that you should discuss more of the effect of pH on calcification in this paper.

Line 26. Add a reference for the hypoxia threshold. Also add the equivalent threshold in $\mu$mol/L (63 $\mu$mol/L) to facilitate the understanding of readers more used to this unit.

L. 33. Remove the successive closing and opening brackets (several occurrences in the whole paper).

L. 42. The paper by Nardelli and coauthors is an experiment in anoxia not hypoxia, specify use as a reference.

L. 53. Provide the species name.

L. 67. Identify that this species lives fixed on hard substrate.

L. 77. "mixture of nitrogen and air" correct to dinitrogen and specify if possible the % ratio of the mixture.

L.79. Specify the stability of the oxygen concentration.

Section 2.1. Specify somewhere how long did the experiment last, and if there were some substrate in the aquariums.

L.87. "2 ml"change to "2 mL" and make sure that this unit is written in capital letters in the whole paper.

Section 2.2 specify if initial size were homogenized or individuals were randomly dispatched in the different DO levels/bottles.

L.105. 5 individuals randomly selected?

L.108. How and where do you measure the initial and final concentrations? Inside or outside of the syringe? In which volume? Do you shake the syringe prior to measurement (or water expelling) to homogenize the water? Is there some stirring inside the syringe? How do you seal the syringe? Could you add some references for this measuring method? If you do not have any water homogenization system, and oxygen concentration is only measured outside of the syringe, there might be some unmeasured "stratification" occurring. There is an important difference in oxygen concentration in between the water close to the syringe tip and the water close to the individuals.

L.110 Give some extra information about the blanks. What is the variation observed in these measurements? Does the hypoxic blank show an increasing oxygen concentration suggesting a potential leak due to the sealing system? In the hypoxic sets of measurements, what was the final oxygen concentration? Did the individuals survive to this oxygen concentration? Respiration rate being size dependent, precisely identify the size distribution of the different samples.

L.122 "+add-on" specify.

L.142. "we demonstrated" at this stage of the stage of the discussion it is too early to assume that you have demonstrated anything.

L.144 and 160. I would recommend avoid using the term "adaptive" due to its potential evolutionary connotation.

L. 154. You mention quite often the "metabolic energy for calcification" is there any way to estimate it and compare it to the loss of energy provided by oxygen respiration? Palmer (1992) gives estimates in terms of J per mass of CaCO3 produced, oxygen respiration rates can also be easily converted to joules (roughly 3000kJ per mole of glucose reduced). With such calculations, you could estimate the part of the respiration-energy required for calcification in both oxygenated and hypoxic conditions.

L. 177. "the anti-predator response is not markedly affected by hypoxia". Do you know who are the predators for this polychaete species and how these predators are sensitive to hypoxia? If the predators are sensitive and/or non-active in hypoxia, there might not be any point of being protected under such conditions.

L. 197. "affecting shell solubility". Precisely how does it affect the solubility (positively or negatively?).

L. 199. "it is predicted that metabolic energy is involved in the control of magnesium incorporation". Is that your prediction or is it some other authors? Please identify the source/reasoning.

L. 211. The citation to your other paper seems irrelevant here since in that paper you do not analyze the effect of hypoxia.

Figure 1 caption. Provide the number of replicates on which your mean and S.E. are based on. Add the initial and final sizes (in Table 1 or supplementary data).

Table 1 caption. Provide the number of replicates on which your mean and S.E. are

based on.

Table S1. This table can appear in the paper and not in the supplementary data. Explain what "MS" stands for. Give the degrees of freedom values.

References list.

L. 254. I would not recommend keeping this citation since this paper was not accepted for publication.

BGD Review criteria:

-Does the paper address relevant scientific questions within the scope of BG? Yes

-Does the paper present novel concepts, ideas, tools, or data? Yes

-Are substantial conclusions reached? Yes, but further discussions are needed

-Are the scientific methods and assumptions valid and clearly outlined? More details about the oxygen respiration method should be provided. The potential effect of pH should be further discussed.

-Are the results sufficient to support the interpretations and conclusions? No, more information about pH should be provided.

-Is the description of experiments and calculations sufficiently complete and precise to allow their reproduction by fellow scientists (traceability of results)? Partially, more information about the respiration measurements should be provided.

Do the authors give proper credit to related work and clearly indicate their own new/original contribution? Yes

-Does the title clearly reflect the contents of the paper? Yes

-Does the abstract provide a concise and complete summary? Yes

-Is the overall presentation well structured and clear? Yes

-Is the language fluent and precise? Yes

-Are mathematical formulae, symbols, abbreviations, and units correctly defined and used? Yes, authors just need to homogenize the writing of "liters" unit.

-Should any parts of the paper (text, formulae, figures, tables) be clarified, reduced, combined, or eliminated? Supplementary table should appear in the paper. The potential effect of other variables than hypoxia (e.g. pH) should be further discussed.

-Are the number and quality of references appropriate? Yes

-Is the amount and quality of supplementary material appropriate? Yes

---

## Referee Comment (RC2) · L.J. de Nooijer (Referee) · 22 May 2017

Dear editor,

I have read the manuscript by Leung and Cheung on the ability to form calcium carbonate of a tubeworm species under anoxic experimental conditions. After careful assessment, I have to advise to reject this manuscript from publication in Biogeosciences. The experimental design is flawed with only two times two treatments (normoxia versus hypoxia and stress versus no-stress) to allow for a sound analysis of the effect of O2 and stress on tube-formation. It is unclear how many polychaetes were incubated

for each treatment, how many survived the experimental period, and exactly how many measurements were done on the formed tubes/ how many analyses were done to determine the respiration, clearance rate, etc. It is a pity that tube length was only assessed once (at the end of the 3-week incubation period) and not throughout the experiment. This would have allowed to estimate whether the worms did not produce their tubes only in e.g. the first days of the experimental period and whether, e.g. absence of an essential nutrient may have hampered growth by these organisms in this experimental setup.

There is no real control group followed to see whether the handling of the specimens in the microcentrifuge tubes affected the worms' functioning. It is unclear how long the incubation were to determine the oxygen uptake (respiration) and whether or not the (under normoxic conditions) O2 levels already decreased towards 0 within the first period of the incubation. Table 1 lists results with averages and their standard error instead the standard deviation. This would only make sense if very many measurements were performed, but that is impossible to tell from this manuscript. It is unclear how measuring DO under hypoxic conditions with this experimental design can result in accurate respiration rates and how they can be compared to rates under normoxia.

The discussion contains many over-interpretations and conclusions have little relation to the results.

Sincerely,

Lennart de Nooijer

---

## Referee Comment (RC3) · Anonymous Referee #3 · 24 May 2017

Review of " Changing mineralogical properties of shells may help minimize the impact of hypoxia-induced metabolic depression on calcification" by Leung and Cheung on Biogeosciences Discussion

There are still limited studies about the influence of oxygen-depleted environment on carbonate shell formation. The reviewer can believe this study of Leung and Cheung give important knowledge on this on-going problem. The contents of consideration are consistent with the interests of the Biogeosciences' audience. On the other hand, there are some remained questions about the results and considerations.

[Figure]

The authors should show the actual sampling/measurement points on the specimens with images. Chan et al. (Journal of Structural Biology 189 (2015) 230-237) reports the shell formation of H. elegans (belonging to the same genus of H. diramphus). According to the study, calcite/aragonite ratio and Mg/Ca altered by ontogeny. Mineralogical heterogeneity was also found in the single shell wall by Chen et al. (2015). The reviewer expects, then, similar variability may be found in H. diramphus, too. The authors seems working with the species for long time. By their knowledge, the ontogenic mineralogical and Mg/Ca variation of H. diramphus should be shown by this study or previous studies. Possibly, the current interpretations are totally revised.

Though authors have considered that calcite/aragonite ratio would depend on energy requirements/consumptions, the ratio is also influenced by magnesium composition and organic compound compositions of mother fluid of calcification, as well as temperature, pressure etc (e.g. Berner, Geochim et Cosmochim Acta 1975, 489-504; Meldrum and Cölfen, Chem. Rev., 2008, 4332-4432). The reviewer expect that the energy for shell formation would be spent on production of the organic compounds of calcification substrate and enzymes, too (ion pumps, carbonic anhydrase etc.). The authors should show the amount of organic matter in the shell. It will be also nice to show the enzyme activity during shell formation for further consideration if possible.

Though the authors thought that the hypoxia condition has increased relative amorphous calcium carbonate (ACC) content by its lower energy demand, according to their cited Weiner and Addadi (1997), it is described that it requires much energy to maintain ACC. There are a discrepancy with authors' consideration.

On the other hand, Fig 1 clearly show authors basic understanding about depression of calcareous shell by low-DO condition. In the hypoxic condition, shell growth slows and hardness increases. This key findings is valuable to publish. It can be understood that this may increase the volume by decreasing the density or the like in order to promote growth in the normal oxygen condition in my mind.

From the above, the result of the authors can be regarded as insufficient to support their own hypothesis. Some additional measurements and restructure of whole manuscript will be necessary to evident authors' current consideration. I would like to recommend resubmission with additional data and discussion.

---

## Referee Comment (RC4) · 24 May 2017

n. keul (Referee)

nkeul@ldeo.columbia.edu

The manuscript " Changing mineralogical properties of shells may help minimize the impact of hypoxia-induced metabolic depression on calcification" by Y.S. Leung and K.M. Cheung describes physiological and shell-compositional responses of a calcifying polychaete to hypoxia and predator stress. The research question as such is original and could potentially provide interesting data to the community. However, the experimental setup and the analytical methods used are inappropriate, at least in the way they are described in the manuscript in the present form. The main points that

need to be addressed:

- 1) stability of DO over the culturing period needs to be reported. Stability in DO could be impacted severely by a) gas-exchange with air (headspace/caps used), b) addition of photosynthetically active algae and c) addition of 1/3 of non treated (e.g. normal DO) seawater every 3 days due to feeding.

2) bubbling with N2 potentially changes also CO2- C-system parameters, also over time, need to be reported

If DO/ C-system were not stable over time, results could be negatively impacted. Furthermore, the data is not presented sufficiently, individual measurements of all parameters need to be reported in a table along with averaged values and standard deviation. These points render the manuscript unfortunately not suitable for publication in its current form. If the authors can address these points, the manuscript could be reconsidered for submission. I wish the author good luck with their resubmission and remain available for further feedback and discussions.

With kind regards, Nina Keul

Main points that need addressing:

-Were any other parameters of the C-system measured in combination with pH to assess stability over time? Bubbling with N2 not only strips oxygen but can also strip CO2. If the two treatments (hypoxia/normoxia) are compared, it needs to be ensured that the C-system was similar, otherwise the effects could be attributed to calcification response to changing C-system (higher carbonate ion conc. caused by higher pH) and not solely to hypoxia effects.

-If I understood correctly, water was exchanged during the culturing period every 3 days. Was the bottle filled headspace free and sealed that air in the headspace on top of the bottle did not equilibrate with the air outside? Was DO conc. in the bottles measured after 3 days to measure stability in DO conc.? Also, the algae provided

photosynthesize, potentially changing DO and pH, was this accounted for? If food was added each day a 20ml, after 3 days, 60ml out of the total of 180 ml (=1/3) does not stem from DO adjusted seawater, so DO concentrations (and pH) could have been significantly different after 3 days.

-Shell growth- How was shell growth measured? Were the individuals Id-ed and growth measured over time or just at the end? Please report shell growth data. How many individuals were cultured? How were the parts of the tube that were added during treatment identified? I assume only parts added in treatment were chosen for analysis of shell properties or was the whole tube analyzed? If analyzing the whole tube, this could potentially mask shell effects on shell properties, as a certain amount of shell would stem from non-treatment conditions. Please also report individual data of shell property measurements.

-Was salinity monitored over the experimental duration? If the air/N2 mix while bubbling was not moist, this could cause salinity to change. Salinity changes also would cause changes in Mg/Ca in the seawater used for culturing, possible causing the Mg/Ca changes reported here (partly). Was Mg/Ca measured in the seawater? In what unit is Mg/Ca reported in Table 1? Mmol/mol?

Minor points: -l. 68- preliminary study: Please provide data- how many organisms were studied, how was survival assessed, how long lasted culturing period, what were experimental conditions (food/temperature, salinity, etc..)?

-measurements of water parameters (l. 73)- what instruments were used? salinity-either unitless or use psu, what pH scale is reported?

l. 133- please report blank values so the reader can assess, how much gas exchange through the syringe occurs over an hour.

---

## Author Comment (AC1) · 30 May 2017

In this paper, J. Leung and N. Cheung describe and discuss the effect of hypoxia on growth, shell properties and chemistry, respiration and feeding activity of Hydroides diramphus. Overall the paper is well written, the data are well presented and discussed but one aspect of the experimental design concerns me. The authors achieved incubation in hypoxia by "aerating seawater with a mixture of nitrogen and air" (l. 76-77). This technique indeed allows replacement of dissolved oxygen by N2 but also tends to remove dissolved CO2 and can then increase water pH. Did the authors measure and

control pH in the different DO treatments? If yes, please provide this information. If no, I'm afraid that most of the discussion on the effect of hypoxia on shell chemistry might be irrelevant giving that changes in pH could explain the changes in calcite/aragonite and Mg/Ca. The lack of discussion of the effect of pH is especially surprising since you mention at least 6 times the ocean acidification in the introduction and none in the discussion. I would therefore recommend to further discuss the potential changes in pH during the experiment (ideally you should measure it) and how it could affect the calcification of this polychaete species.

RESPONSE: We can provide the pH data. The reviewer is right to state that pH was slightly elevated by pumping the mixture of nitrogen gas and air into the seawater. How basification of seawater affects Mg/Ca and calcite/aragonite remains largely unexplored, but can be inferred from studies on ocean acidification. In the context of ocean acidification, Mg/Ca in calcite would be reduced as high-Mg calcite is more susceptible to dissolution. Yet, the huge difference in Mg/Ca (hypoxia vs. normoxia) is unlikely only caused by the small difference in pH. Since Mg/Ca is mainly under biological control (Bentov and Erez, 2006), we suggest that hypoxia is the major factor causing the difference in Mg/Ca via its impact on energy metabolism. Calcite/aragonite could be altered in bimineralic calcifiers under ocean acidification (usually higher proportion of calcite) because aragonite will become limiting first due to higher solubility than calcite (Feely et al., 2004). However, this prediction cannot be applied to basification because both calcite and aragonite are saturated. We will discuss the potential pH effect in the revision as requested.

To summarize, I would recommend the publication of the present paper in the following conditions: -the authors detail if/how they controlled the pH during the experiment; -estimate or measure the pH changes during N2 bubbling; - discuss of how could pH changes (if existing) could explain together with hypoxia the shell chemistry observations you made.

RESPONSE: We will address these points in the revision.

In addition, here are some extra comments on the manuscript: Introduction section. In the introduction, you mention at least 6 times the ocean acidification, a phenomenon that you do not mention again at any point later in the paper. As explained above I think that you should discuss more of the effect of pH on calcification in this paper.

RESPONSE: We mentioned ocean acidification to highlight that calcification is very likely not related to carbonate chemistry in seawater according to recent studies; therefore, we suggested that calcification is mainly associated with physiological performance, which would be greatly impaired by hypoxia. We agree that ocean acidification was a bit overemphasised in the Introduction and will be tuned down in the revision.

Line 26. Add a reference for the hypoxia threshold. Also add the equivalent threshold in $\mu$mol/L (63 $\mu$mol/L) to facilitate the understanding of readers more used to this unit.

RESPONSE: Suggestions will be adopted in the revision.

L. 33. Remove the successive closing and opening brackets (several occurrences in the whole paper).

RESPONSE: Suggestion will be adopted in the revision.

L. 42. The paper by Nardelli and coauthors is an experiment in anoxia not hypoxia, specify use as a reference.

RESPONSE: The sentence will be revised as ". . . calcification under hypoxia (Mukherjee et al., 2013; Keppel et al., 2016), and even anoxia (Nardelli et al., 2014)".

L. 53. Provide the species name.

RESPONSE: Species name will be added in the revision.

L. 67. Identify that this species lives fixed on hard substrate.

RESPONSE: This information will be added in the revision.

L. 77. "mixture of nitrogen and air" correct to dinitrogen and specify if possible the %

ratio of the mixture.

RESPONSE: Suggestions will be adopted, except that we prefer to use "nitrogen gas" rather than "dinitrogen" as more readers can understand the former.

L.79. Specify the stability of the oxygen concentration.

RESPONSE: We will add a table to show the stability of O2 concentration.

Section 2.1. Specify somewhere how long did the experiment last, and if there were some substrate in the aquariums.

RESPONSE: The experiment lasted for 3 weeks and there was no substrate in the aquarium.

L.87. "2 ml"change to "2 mL" and make sure that this unit is written in capital letters in the whole paper.

RESPONSE: We will check and correct this unit throughout the manuscript.

Section 2.2 specify if initial size were homogenized or individuals were randomly dispatched in the different DO levels/bottles.

RESPONSE: Before the exposure, we standardized the size of individuals which were then randomly assigned to the bottles. This information will be added in the revision.

L.105. 5 individuals randomly selected?

RESPONSE: Yes. Information will be added in the revision.

L.108. How and where do you measure the initial and final concentrations? Inside or outside of the syringe? In which volume? Do you shake the syringe prior to measurement (or water expelling) to homogenize the water? Is there some stirring inside the syringe? How do you seal the syringe? Could you add some references for this measuring method? If you do not have any water homogenization system, and oxygen concentration is only measured outside of the syringe, there might be some unmeasured

"stratification" occurring. There is an important difference in oxygen concentration in between the water close to the syringe tip and the water close to the individuals.

RESPONSE: The initial and final DO concentrations were measured by inserting the DO probe into the tip of syringe which was sealed by Blu-Tack after initial measurement. The volume of seawater in the syringe was $\sim$35 mL (Ln 129). For each measurement, we have to ensure that the DO concentration is uniform by gently stirring the seawater with the DO probe, which provides the real-time measurement of DO concentration. The reading was taken when the DO concentration is stable along the depth to ensure no "stratification". It is not advised to constantly stir the seawater inside the syringe, which causes stress to the individuals. Relevant references are Zhao et al. (2011) and Leung et al. (2013). We will add more information in the revision for clarity.

L.110 Give some extra information about the blanks. What is the variation observed in these measurements? Does the hypoxic blank show an increasing oxygen concentration suggesting a potential leak due to the sealing system? In the hypoxic sets of measurements, what was the final oxygen concentration? Did the individuals survive to this oxygen concentration? Respiration rate being size dependent, precisely identify the size distribution of the different samples.

RESPONSE: The blank is used to correct the background change in DO concentration, which was negligible after the 1-hour exposure, including the hypoxic blank ($\pm$ $\sim$1%). This indicates that the syringe was sealed properly. The final measurement for the hypoxic group was about 0.5 mg L-1 and H. diramphus was able to survive after the measurement due to strong tolerance to hypoxia. Before the exposure, we standardized the body size to about 20 mm and randomly assigned the individuals to each bottle to avoid bias. Information will be added in the revision.

L.122 "+add-on" specify.

RESPONSE: "PERMANOVA+ add-on" is the name of software.

L.142. "we demonstrated" at this stage of the stage of the discussion it is too early to assume that you have demonstrated anything.

RESPONSE: We will change "demonstrated" to "found".

L.144 and 160. I would recommend avoid using the term "adaptive" due to its potential evolutionary connotation.

RESPONSE: We will change "adaptive" to "beneficial, favourable, positive, etc.".

L. 154. You mention quite often the "metabolic energy for calcification" is there any way to estimate it and compare it to the loss of energy provided by oxygen respiration? Palmer (1992) gives estimates in terms of J per mass of CaCO3 produced, oxygen respiration rates can also be easily converted to joules (roughly 3000kJ per mole of glucose reduced). With such calculations, you could estimate the part of the respiration-energy required for calcification in both oxygenated and hypoxic conditions.

RESPONSE: We appreciate this comment. In this study, however, it is premature to estimate the actual metabolic cost of calcification as we did not measure the weight of CaCO3 produced and ingredients of the shell (e.g. proteins and carbohydrates), which are species-specific. The energetics of calcification is beyond the scope of this study. Studying energetics is challenging (to us) since there are many parameters should be considered. Palmer's model is good, but more parameters can still be added to provide an accurate estimation (see dynamic energy budget model).

L. 177. "the anti-predator response is not markedly affected by hypoxia". Do you know who are the predators for this polychaete species and how these predators are sensitive to hypoxia? If the predators are sensitive and/or non-active in hypoxia, there might not be any point of being protected under such conditions.

RESPONSE: Predators are predatory fish, such as spinefoots. Yet, humans are responsible for the shell damage during the removal of H. diramphus from many man-made structures, which can happen irrespective of DO concentration. Anti-predator

response is another topic in ecology and therefore we will change "anti-predator response" to "defensive response" or similar wordings to avoid confusing readers.

L. 197. "affecting shell solubility". Precisely how does it affect the solubility (positively or negatively?).

RESPONSE: We will change "affecting" to "reducing" or other similar words.

L. 199. "it is predicted that metabolic energy is involved in the control of magnesium incorporation". Is that your prediction or is it some other authors? Please identify the source/reasoning.

RESPONSE: It is our prediction but a previous study suggests that energy is involved in the control of magnesium incorporation (Bentov and Erez, 2006). We will elaborate in the revision.

L. 211. The citation to your other paper seems irrelevant here since in that paper you do not analyze the effect of hypoxia.

RESPONSE: Although the stressor is different, the concept in this paper is highly relevant and thus we wrote "see also" just to recommend reading another similar study.

Figure 1 caption. Provide the number of replicates on which your mean and S.E. are based on. Add the initial and final sizes (in Table 1 or supplementary data).

RESPONSE: Information will be added in the revision.

Table 1 caption. Provide the number of replicates on which your mean and S.E. are based on.

RESPONSE: Information will be added in the revision.

Table S1. This table can appear in the paper and not in the supplementary data.

RESPONSE: Suggestion will be adopted in the revision.

Explain what "MS" stands for. Give the degrees of freedom values.

RESPONSE: We will change MS to "Mean square" and add df values in the revision.

References list. L. 254. I would not recommend keeping this citation since this paper was not accepted for publication.

RESPONSE: Even though this paper is not accepted, it has still been highly cited. We will decide whether this citation should be kept.

BGD Review criteria:

-Does the paper address relevant scientific questions within the scope of BG? Yes

-Does the paper present novel concepts, ideas, tools, or data? Yes

-Are substantial conclusions reached? Yes, but further discussions are needed

RESPONSE: We will further discuss the findings as suggested.

-Are the scientific methods and assumptions valid and clearly outlined? More details about the oxygen respiration method should be provided. The potential effect of pH should be further discussed.

RESPONSE: We will describe the O2 measurement clearly as requested and discuss the potential pH effect.

-Are the results sufficient to support the interpretations and conclusions? No, more information about pH should be provided.

RESPONSE: We will provide pH data in the revision.

-Is the description of experiments and calculations sufficiently complete and precise to allow their reproduction by fellow scientists (traceability of results)? Partially, more information about the respiration measurements should be provided.

RESPONSE: More information will be provided in the revision.

Do the authors give proper credit to related work and clearly indicate their own

new/original contribution? Yes

-Does the title clearly reflect the contents of the paper? Yes

-Does the abstract provide a concise and complete summary? Yes

-Is the overall presentation well structured and clear? Yes

-Is the language fluent and precise? Yes

-Are mathematical formulae, symbols, abbreviations, and units correctly defined and used? Yes, authors just need to homogenize the writing of "liters" unit.

RESPONSE: We will make this unit consistent in the revision.

-Should any parts of the paper (text, formulae, figures, tables) be clarified, reduced, combined, or eliminated? Supplementary table should appear in the paper. The potential effect of other variables than hypoxia (e.g. pH) should be further discussed.

RESPONSE: Suggestions will be adopted in the revision.

-Are the number and quality of references appropriate? Yes

-Is the amount and quality of supplementary material appropriate? Yes

References

Bentov S and Erez J (2006) Impact of biomineralization processes on the Mg content of foraminiferal shells: A biological perspective. Geochem. Geophys. Geosyst. 7, Q01P08.

Feely RA et al. (2004) Impact of anthropogenic $CO_2$ on the $CaCO_3$ system in the oceans. Science 305, 362–366.

Leung YS et al. (2013) Physiological and behavioural responses of different life stages of a serpulid polychaete to hypoxia. Mar. Ecol. Prog. Ser. 477, 135–145.

Zhao Q et al. (2011) Effects of starvation on the physiology and foraging behaviour of

two subtidal nassariid scavengers. J. Exp. Mar. Biol. Ecol. 409, 53–61.

---

## Author Comment (AC2) · 31 May 2017

The manuscript " Changing mineralogical properties of shells may help minimize the impact of hypoxia-induced metabolic depression on calcification" by Y.S. Leung and K.M. Cheung describes physiological and shell-compositional responses of a calcifying polychaete to hypoxia and predator stress. The research question as such is original and could potentially provide interesting data to the community. However, the experimental setup and the analytical methods used are inappropriate, at least in the way they are described in the manuscript in the present form. The main points that

need to be addressed:

- 1) stability of DO over the culturing period needs to be reported. Stability in DO could be impacted severely by a) gas-exchange with air (headspace/caps used), b) addition of photosynthetically active algae and c) addition of 1/3 of non treated (e.g. normal DO) seawater every 3 days due to feeding.

2) bubbling with N2 potentially changes also CO2- C-system parameters, also over time, need to be reported

If DO/ C-system were not stable over time, results could be negatively impacted. Furthermore, the data is not presented sufficiently, individual measurements of all parameters need to be reported in a table along with averaged values and standard deviation. These points render the manuscript unfortunately not suitable for publication in its current form. If the authors can address these points, the manuscript could be reconsidered for submission. I wish the author good luck with their resubmission and remain available for further feedback and discussions.

RESPONSE: We are sorry for excluding seawater data in the previous submission as we thought that they have limited interpretive values. Seawater data will be reported in the revision as requested. We appreciate reviewer's blessing for the resubmission.

Main points that need addressing:

-Were any other parameters of the C-system measured in combination with pH to assess stability over time? Bubbling with N2 not only strips oxygen but can also strip CO2. If the two treatments (hypoxia/normoxia) are compared, it needs to be ensured that the C-system was similar, otherwise the effects could be attributed to calcification response to changing C-system (higher carbonate ion conc. caused by higher pH) and not solely to hypoxia effects.

RESPONSE: This comment is also raised by Reviewer 1. We will add seawater data and discuss the potential effect of basification in the revision.

-If I understood correctly, water was exchanged during the culturing period every 3 days. Was the bottle filled headspace free and sealed that air in the headspace on top of the bottle did not equilibrate with the air outside? Was DO conc. in the bottles measured after 3 days to measure stability in DO conc.? Also, the algae provided photosynthesize, potentially changing DO and pH, was this accounted for? If food was added each day a 20ml, after 3 days, 60ml out of the total of 180 ml (=1/3) does not stem from DO adjusted seawater, so DO concentrations (and pH) could have been significantly different after 3 days.

RESPONSE: The glass bottle was covered with a lid to prevent interaction with air outside. Two holes were drilled on the lid: one for inserting an airline to supply the gases continuously and one for equalizing the pressure inside and outside the bottle. This information will be added in the revision for clarity. DO concentration was monitored regularly throughout the 3-week exposure period. We need to emphasize that the effect of respiration and photosynthesis on DO concentration was negligible because the seawater in the experimental setup was continuously aerated so that stable equilibrium of gases can be achieved throughout the experiment (Ln 76-80). As such, the minimally increased DO concentration due to addition of algal suspension can be returned to the desired DO level rapidly (i.e. negative feedback).

-Shell growth- How was shell growth measured? Were the individuals Id-ed and growth measured over time or just at the end? Please report shell growth data. How many individuals were cultured? How were the parts of the tube that were added during treatment identified? I assume only parts added in treatment were chosen for analysis of shell properties or was the whole tube analyzed? If analyzing the whole tube, this could potentially mask shell effects on shell properties, as a certain amount of shell would stem from non-treatment conditions. Please also report individual data of shell property measurements.

RESPONSE: Shell growth was estimated by measuring tube length (Ln 95), while shell growth rate is given by (final tube length – initial tube length)/exposure time (Leung and

Cheung, 2017). Ten individuals were cultured in each bottle (Ln 89). Microcentrifuge tubes were labelled to identify each individual (one tube, one individual). We measured tube length three times (before, in the mid of and after the exposure), but overall shell growth rate has the greatest interpretive value. The shell growth data can be provided as supplementary information. As for the analysis of shell properties, we only used newly-produced shells (Ln 96-97) because the properties of old shells are probably unchanged. It is very easy to identify the newly-produced shell. Photos will be provided in the revision for illustration.

-Was salinity monitored over the experimental duration? If the air/N2 mix while bubbling was not moist, this could cause salinity to change. Salinity changes also would cause changes in Mg/Ca in the seawater used for culturing, possible causing the Mg/Ca changes reported here (partly). Was Mg/Ca measured in the seawater? In what unit is Mg/Ca reported in Table 1? Mmol/mol?

RESPONSE: Salinity was checked regularly throughout as H. diramphus is relatively sensitive to salinity change. Salinity was very stable because seawater was renewed once every three days and seawater evaporation is minimal in the bottle with a lid. Unfortunately, we did not measure Mg/Ca of seawater, which has a stable value on a large geographic scale. Regardless, we used the same bulk of seawater across treatments so that no bias was induced. Mg/Ca is a molar ratio and we will add "(molar)" after "Mg/Ca in calcite" to match Ries's presentation, which is common in climate change research (e.g. Ries, 2010).

Minor points: -l. 68- preliminary study: Please provide data- how many organisms were studied, how was survival assessed, how long lasted culturing period, what were experimental conditions (food/temperature, salinity, etc..)?

RESPONSE: The method was described in our previous study (Leung et al., 2013), except that different species was used. After careful consideration, we decided to remove this redundant sentence as polychaetes are generally regarded to have strong

tolerance to hypoxia (Vaquer-Sunyer and Duarte, 2008). We will provide the survival rate following exposure to show the tolerance of H. diramphus to hypoxia. This can substantiate that H. diramphus is a suitable species for this study, while avoiding detailed description of the unpublished data in the preliminary study, which is not very relevant to this study.

-measurements of water parameters (l. 73)- what instruments were used? Salinity either unitless or use psu, what pH scale is reported?

RESPONSE: Information on instruments (e.g. pH meter, refractometer, etc.) used for each water parameter will be added in the revision. NBS scale was reported.

l. 133- please report blank values so the reader can assess, how much gas exchange through the syringe occurs over an hour.

RESPONSE: Suggestion will be adopted in the revision.

References

Leung JYS, Cheung NKM (2017) Feeding behaviour of a serpulid polychaete: Turning a nuisance species into a natural resource to counter algal blooms? Mar. Pollut. Bull. 115, 379–382.

Leung YS, Shin PKS, Qiu JW, Ang PO, Chiu JMY, Thiyagarajan V, Cheung SG (2013) Physiological and behavioural responses of different life stages of a serpulid polychaete to hypoxia. Mar. Ecol. Prog. Ser. 477, 135–145.

Ries JB (2010) Review: geological and experimental evidence for secular variation in seawater Mg/Ca (calcite-aragonite seas) and its effects on marine biological calcification. Biogeosciences 7, 2795–2849.

Vaquer-Sunyer R, Duarte CM (2008) Thresholds of hypoxia for marine biodiversity. Proc. Natl. Acad. Sci. 105, 15452–15457.

---

## Referee Comment (RC5) · Anonymous Referee #5 · 1 Jun 2017

The manuscript "Changing mineralogical properties of shells may help minimize the impact of hypoxia-induced metabolic depression on calcification" presents a study on the influence of Hypoxia on respiration rates, shell growth and clearance rate of the calcifying polycheate Hydroides diramphus. Considering the the trend of expanding OMZs in the modern ocean, studies on the influence of Hypoxia on different organisms is getting more and more important.

Since there are already 4 detailed reviews presents for the manuscript I try to keep it short. Most of my issues have already been addressed by the other reviewers. My

main concerns also include the tracking of pH and oxygen during the incubations. The authors already responded that they will provide the pH for the experiment and to discuss the possible influence of the pH change of the organism. It is still not clear to me, though, if and how oxygen was controlled and the Hypoxia were maintained. Was the gas mixture bubbled through the bottles which were used for the incubations during the experiment or was just hypoxic water added to the bottles and then they were closed? In the latter case I would think the water wouldn't stay hypoxic very long. Was the oxygenation constantly monitored or just during certain times of the experiment? I think this is the most urgent information which has to be provided/clarified in the revised manuscript. Furthermore, I think the manuscript would benefit if parts of the discussion which include some overinterpretations or generalisations would be rewritten.

―――――――――――――――――――

---

## Author Comment (AC4) · 1 Jun 2017

There are still limited studies about the influence of oxygen-depleted environment on carbonate shell formation. The reviewer can believe this study of Leung and Cheung give important knowledge on this on-going problem. The contents of consideration are consistent with the interests of the Biogeosciences' audience.

RESPONSE: Thank you very much for your interest in our manuscript.

On the other hand, there are some remained questions about the results and consid-

erations. The authors should show the actual sampling/measurement points on the specimens with images. Chan et al. (Journal of Structural Biology 189 (2015) 230-237) reports the shell formation of H. elegans (belonging to the same genus of H. diramphus). According to the study, calcite/aragonite ratio and Mg/Ca altered by ontogeny. Mineralogical heterogeneity was also found in the single shell wall by Chen et al. (2015). The reviewer expects, then, similar variability may be found in H. diramphus, too. The authors seems working with the species for long time. By their knowledge, the ontogenic mineralogical and Mg/Ca variation of H. diramphus should be shown by this study or previous studies. Possibly, the current interpretations are totally revised.

RESPONSE: We appreciate this comment. As highlighted in the Introduction and Title, however, this study aims to examine whether calcifying organisms can modify mineralogical properties (i.e. mineralogical plasticity) to possibly accommodate hypoxia, which has great research interest and novelty in physiological ecology. To answer this ecological question, mineralogical properties and physiological performance are needed. Technically, we used the whole newly-produced shells for the analysis of mineralogical properties (Ln 96-97), which can show the overall change in mineralogy that we are interested in. The developmental process of shells (or ontogeny) is beyond the scope of this study, but we can provide SEM images of shell ultrastructure to show whether there are significant changes in the morphology or density of carbonate crystals among treatments, which may be associated with mechanical strength.

Though authors have considered that calcite/aragonite ratio would depend on energy requirements/consumptions, the ratio is also influenced by magnesium composition and organic compound compositions of mother fluid of calcification, as well as temperature, pressure etc (e.g. Berner, Geochim et Cosmochim Acta 1975, 489-504; Meldrum and Cölfen, Chem. Rev., 2008, 4332-4432). The reviewer expect that the energy for shell formation would be spent on production of the organic compounds of calcification substrate and enzymes, too (ion pumps, carbonic anhydrase etc.). The authors should show the amount of organic matter in the shell.

[Figure]

RESPONSE: We also recognize that some environmental factors can affect mineralogical properties. In this study, however, we only manipulated two factors (dissolved oxygen and shell damage), while other factors remained unchanged, meaning that the change in mineralogical properties is caused by these two factors (i.e. sources of variation). Organic matter content in the shell is highly related to mechanical properties; therefore, we analysed it few days ago as requested, and the result will be added in the revision.

It will be also nice to show the enzyme activity during shell formation for further consideration if possible.

RESPONSE: Enzymatic activity (e.g. carbonic anhydrase) or other molecular responses during shell formation are beyond the scope of this study, but can be examined with ontogeny in future investigation to elucidate the underlying mechanisms causing different shell properties or growth.

Though the authors thought that the hypoxia condition has increased relative amorphous calcium carbonate (ACC) content by its lower energy demand, according to their cited Weiner and Addadi (1997), it is described that it requires much energy to maintain ACC. There are a discrepancy with authors' consideration.

RESPONSE: The reviewer might have misunderstood our idea. We first need to recognize that stabilization of ACC requires metabolic energy, regardless of DO concentration (i.e. same energy cost between normoxia and hypoxia). Yet, crystallization of ACC to form calcite or aragonite requires extra metabolic energy. The higher relative ACC content indicates that the animals were reluctant/unable to allocate extra metabolic energy to crystallization, possibly due to the reduced energy budget under hypoxia.

On the other hand, Fig 1 clearly show authors basic understanding about depression of calcareous shell by low-DO condition. In the hypoxic condition, shell growth slows and hardness increases. This key findings is valuable to publish. It can be understood that this may increase the volume by decreasing the density or the like in order to promote

growth in the normal oxygen condition in my mind.

RESPONSE: We appreciate reviewer's attempt to interpret the finding. The reduced shell growth under hypoxia is not surprising as it has been previously demonstrated. The most exciting finding is the maintenance of rapid shell growth plus production of harder shells under hypoxia, when the shell was damaged. This unexpected finding suggests that H. diramphus is tolerant to hypoxia and able to allocate more energy to the defensive response, despite the weakened aerobic metabolism. This response is vital for survival.

From the above, the result of the authors can be regarded as insufficient to support their own hypothesis. Some additional measurements and restructure of whole manuscript will be necessary to evident authors' current consideration. I would like to recommend resubmission with additional data and discussion.

RESPONSE: This reviewer is probably a specialist in structural biology and wants to see more technical details. Physiological ecology, however, concerns how environmental stressors impact the fitness and survival of organisms and whether/how organisms can adapt to altered conditions. As such, studies on climate change biology are mainly based on the overall response of organisms to stressors (please see some of the ecological studies below). After careful consideration, we will provide SEM images and organic matter content data in the revision, which are relevant and can greatly strengthen this manuscript.

References

Cheung, S.G., Chan, H.Y., Liu, C.C., and Shin, P.K.S.: Effect of prolonged hypoxia on food consumption, respiration, growth and reproduction in marine scavenging gastropod Nassarius festivus. Mar. Pollut. Bull., 57, 280–286, 2008.

Hirsch, P.E., Cayon, D., and Svanbäck, R.: Plastic responses of a sessile prey to multiple predators: a field and experimental study. PLOS ONE, 9, e115192, 2014.

Leung, J.Y.S., Russell, B.D., and Connell, S.D.: Mineralogical plasticity acts as a compensatory mechanism to the impacts of ocean acidification. Environ. Sci. Technol., 51, 2652–2659, 2017.

Ramajo, L., Rodríguez-Navarro, A.B., Duarte, C.M., Lardies, M.A., and Lagos, N.A.: Shifts in shell mineralogy and metabolism of Concholepas concholepas juveniles along the Chilean coast. Mar. Freshw. Res., 66, 1147–1157, 2015.

Ries, J.B., Cohen, A.L., and McCorkle, D.C.: Marine calcifiers exhibit mixed responses to CO2-induced ocean acidification. Geology, 37, 1131–1134, 2009.

Trussell, G.C. and Nicklin, M.O.: Cue sensitivity, inducible defense, and tradeoffs in a marine snail. Ecology, 83, 1635–1647, 2002.
* * *

---

## Author Comment (AC5) · 2 Jun 2017

The manuscript "Changing mineralogical properties of shells may help minimize the impact of hypoxia-induced metabolic depression on calcification" presents a study on the influence of Hypoxia on respiration rates, shell growth and clearance rate of the calcifying polycheate Hydroides diramphus. Considering the the trend of expanding OMZs in the modern ocean, studies on the influence of Hypoxia on different organisms is getting more and more important.

RESPONSE: We appreciate reviewer's understanding of the importance of this study.

Since there are already 4 detailed reviews presents for the manuscript I try to keep it short. Most of my issues have already been addressed by the other reviewers. My main concerns also include the tracking of pH and oxygen during the incubations. The authors already responded that they will provide the pH for the experiment and to discuss the possible influence of the pH change of the organism. It is still not clear to me, though, if and how oxygen was controlled and the Hypoxia were maintained. Was the gas mixture bubbled through the bottles which were used for the incubations during the experiment or was just hypoxic water added to the bottles and then they were closed? In the latter case I would think the water wouldn't stay hypoxic very long. Was the oxygenation constantly monitored or just during certain times of the experiment? I think this is the most urgent information which has to be provided/clarified in the revised manuscript. Furthermore, I think the manuscript would benefit if parts of the discussion which include some overinterpretations or generalisations would be rewritten.

RESPONSE: The desired dissolved oxygen (DO) concentration was controlled by continuously bubbling N2 and air mixture into the seawater of the experimental setup, which allows equilibrium of gases to achieve constantly. When the equilibrium of gases is achieved, the DO concentration becomes very stable over time as long as the flow rate of gases is maintained. This method for manipulating DO concentration has been widely applied in hypoxia studies (please see some examples below), in addition to our published hypoxia studies. The DO concentration was constantly monitored and recorded daily. We will add more information in the revision for clarity and discuss the potential pH effect to strengthen the discussion.

References

Cheung SG et al. (2008) Effect of prolonged hypoxia on food consumption, respiration, growth and reproduction in marine scavenging gastropod Nassarius festivus. Mar. Pollut. Bull. 57, 280–286.

Mukherjee J et al. (2013) Proteomic response of marine invertebrate larvae to ocean acidification and hypoxia during metamorphosis and calcification. J. Exp. Biol. 213, 4580–4589.

Shang E and Wu R (2004) Aquatic hypoxia is a teratogen and affects fish embryonic development. Environ. Sci. Technol. 38, 4763–4767.

Wang S et al. (2016) Hypoxia causes transgenerational impairments in reproduction of fish. Nat. Commun. 7, 12114.

Wu R et al. (2003) Aquatic hypoxia is an endocrine disruptor and impairs fish reproduction. Environ. Sci. Technol. 37, 1137–1141.

---

## Author Comment (AC6) · 2 Jun 2017

COMMENT 1: I have read the manuscript by Leung and Cheung on the ability to form calcium carbonate of a tubeworm species under anoxic experimental conditions. After careful assessment, I have to advise to reject this manuscript from publication in Biogeosciences. The experimental design is flawed with only two times two treatments (normoxia versus hypoxia and stress versus no-stress) to allow for a sound analysis of the effect of O2 and stress on tube-formation.

[Figure]

RESPONSE 1: We completely disagree that 2 x 2 factorial design is an experimental flaw. To evaluate the effects of hypoxia and shell damage, two groups for each factor are statistically enough. Thus, our full factorial design can provide a sound interpretation for the effects of hypoxia, shell damage and their interaction. In fact, 2 x 2 factorial design is extremely common in ecological studies (e.g. Connell and Russell, 2010; Mukherjee et al., 2013; Ghedini et al., 2015).

COMMENT 2: It is unclear how many polychaetes were incubated for each treatment, how many survived the experimental period, and exactly how many measurements were done on the formed tubes/ how many analyses were done to determine the respiration, clearance rate, etc.

RESPONSE 2: For incubation, we have 10 individuals per bottle (Ln 89) and 3 replicate bottles per DO level per context (Ln 92). For shell hardness and elasticity, we have 5 fragments from 5 individuals per DO level per context (Ln 100-101). We have 3 replicates for calcite to aragonite ratio (Ln 108-109), magnesium to calcium ratio (Ln 117) and amorphous calcium carbonate (Ln 124-125). We have 5 replicate syringes per DO level per context for respiration rate (Ln 129-130) and 5 replicate bottles per DO level per context for clearance rate (Ln 136). The number of replicates for each parameter has been clearly written. We can add survival rate as supplementary information in the revision as it is not very relevant to the hypothesis of this study.

COMMENT 3: It is a pity that tube length was only assessed once (at the end of the 3-week incubation period) and not throughout the experiment. This would have allowed to estimate whether the worms did not produce their tubes only in e.g. the first days of the experimental period and whether, e.g. absence of an essential nutrient may have hampered growth by these organisms in this experimental setup.

RESPONSE 3: We measured tube length three times (before, in the mid of and after the exposure), but the overall shell growth rate has the greatest interpretive value and should be presented. This study does not aim to examine whether the individuals produce tubes on the first few days or not, which has limited scientific value. Nutrition is not a problem as the food, which was provided daily (Ln 92-93), is optimal for supporting tube growth (Leung and Cheung, 2017).

COMMENT 4: There is no real control group followed to see whether the handling of the specimens in the microcentrifuge tubes affected the worms' functioning.

RESPONSE 4: We are not interested in studying the effect of containers on physiological functioning. Microcentrifuge tubes are used to allow the individuals to grow in the upright position and avoid individual interaction (Leung and Cheung, 2017).

COMMENT 5: It is unclear how long the incubation were to determine the oxygen uptake (respiration) and whether or not the (under normoxic conditions) O2 levels already decreased towards 0 within the first period of the incubation.

RESPONSE 5: The incubation time was 1 hour (Ln 131). The speculation that "whether or not the (under normoxic conditions) O2 levels already decreased towards 0 within the first period of the incubation" is ungrounded. The animals were incubated inside a sealed syringe for a single time only; therefore, there is no first or second period of incubation. From normoxia to almost zero DO concentration in 1 hour, the respiration rate needs to be $\sim$42 $\mu$g O2 ind-1 hr-1, which is much higher than what we reported ($\sim$26 $\mu$g O2 ind-1 hr-1).

COMMENT 6: Table 1 lists results with averages and their standard error instead the standard deviation. This would only make sense if very many measurements were performed, but that is impossible to tell from this manuscript.

RESPONSE 6: The choice of standard error (of mean) over standard deviation is required by the statistical context. The core message of Table 1 is how hypoxia and shell damage affect physiological performance and shell geochemical properties. This can only be shown by inferential statistics that highlight and quantify the difference between treatment and control groups, i.e. estimated central tendency and the error of the estimation (e.g. mean $\pm$ SE). The use of plain descriptive statistics, such as SD, is not justifiable in this case. If the readers (out of curiosity or whatever reasons) would like to know how the individuals from the same treatment group differ from each other, SD can be easily calculated by multiplying the provided SE by the square root of sample size (see RESPONSE 2). In addition, the statement "This (SE) would only make sense if very many measurements were performed" is logically and mathematically incorrect. Given that SE = SD/sqrt(N), where N is the sample size, SE will approach zero if N is a very large number. On top of that, one has to be very careful with inferential statistics that come from a very large sample size as tiny difference between the estimates of central tendency of treatment groups will manifest as "significant difference" in statistical sense, but not necessarily in biological sense. Therefore, showing SE can be misleading if N is very large.

COMMENT 7: It is unclear how measuring DO under hypoxic conditions with this experimental design can result in accurate respiration rates and how they can be compared to rates under normoxia.

RESPONSE 7: This experimental design is widely used for comparison across DO treatments (e.g. Zhao et al., 2011; Leung et al., 2013).

COMMENT 8: The discussion contains many over-interpretations and conclusions have little relation to the results.

RESPONSE 8: This comment is ambiguous. The discussion and conclusion are based on our results and hypotheses.

OVERALL RESPONSE: While we are pleased to accept comments, it is also our responsibility to validate them. In this case, many details are substantially overlooked by this reviewer (e.g. COMMENTS 2 & 5) and some comments are subjective without clear reasons (e.g. COMMENTS 1, 7 & 8). If rejection is based on the careless assessment, the quality and objectivity of this review become questionable.

[Figure]

References

Connell SD, Russell BD (2010) The direct effects of increasing CO2 and temperature on non-calcifying organisms: increasing the potential for phase shifts in kelp forests. Proc. R. Soc. B, 277, 1409–1415.

Ghedini G, Russell BD, Connell SD (2015) Trophic compensation reinforces resistance: herbivory absorbs the increasing effects of multiple disturbances. Ecol. Lett. 18, 182–187.

Leung YS, Shin PKS, Qiu JW, Ang PO, Chiu JMY, Thiyagarajan V, Cheung, SG (2013) Physiological and behavioural responses of different life stages of a serpulid polychaete to hypoxia. Mar. Ecol. Prog. Ser. 477, 135–145.

Leung JYS, Cheung NKM (2017) Feeding behaviour of a serpulid polychaete: Turning a nuisance species into a natural resource to counter algal blooms? Mar. Pollut. Bull. 115, 379–382.

Mukherjee J, Wong KKW, Chandramouli KH, Qian PY, Leung PTY, Wu RSS, Thiyagarajan V (2013) Proteomic response of marine invertebrate larvae to ocean acidification and hypoxia during metamorphosis and calcification. J. Exp. Biol. 213, 4580–4589.

Zhao Q, Cheung SG, Shin PKS, Chiu, JMY (2011) Effects of starvation on the physiology and foraging behaviour of two subtidal nassariid scavengers. J. Exp. Mar. Biol. Ecol. 409, 53–61.